# Kynurenine Pathway Metabolites as Potential Biomarkers in Chronic Pain

**DOI:** 10.3390/ph16050681

**Published:** 2023-05-02

**Authors:** Andrew Auyeung, Hank C. Wang, Kannan Aravagiri, Nebojsa Nick Knezevic

**Affiliations:** 1Advocate Illinois Masonic Medical Center, Department of Anesthesiology, Chicago, IL 60657, USA; andrew.w.auyeung@dmu.edu (A.A.);; 2College of Osteopathic Medicine, Des Moines University, Des Moines, IA 50312, USA; 3Chicago Medical School, Rosalind Franklin University of Medicine and Science, North Chicago, IL 60064, USA; 4Department of Anesthesiology, University of Illinois, Chicago, IL 60612, USA; 5Department of Surgery, University of Illinois, Chicago, IL 60612, USA

**Keywords:** chronic pain, biomarkers, kynurenine pathway, quinolinic acid, kynurenic acid, tryptophan metabolism, inflammation, neurotoxicity

## Abstract

Chronic pain is a pressing medical and socioeconomic issue worldwide. It is debilitating for individual patients and places a major burden on society in the forms of direct medical costs and lost work productivity. Various biochemical pathways have been explored to explain the pathophysiology of chronic pain in order to identify biomarkers that can potentially serve as both evaluators of and guides for therapeutic effectiveness. The kynurenine pathway has recently been a source of interest due to its suspected role in the development and sustainment of chronic pain conditions. The kynurenine pathway is the primary pathway responsible for the metabolization of tryptophan and generates nicotinamide adenine dinucleotide (NAD^+^), in addition to the metabolites kynurenine (KYN), kynurenic acid (KA), and quinolinic acid (QA). Dysregulation of this pathway and changes in the ratios of these metabolites have been associated with numerous neurotoxic and inflammatory states, many of which present simultaneously with chronic pain symptoms. While further studies utilizing biomarkers to elucidate the kynurenine pathway’s role in chronic pain are needed, the metabolites and receptors involved in its processes nevertheless present researchers with promising sources of novel and personalized disease-modifying treatments.

## 1. Introduction

Chronic pain is characterized as pain that persists for over twelve weeks despite medication or treatment [1]. Chronic pain may arise due to a multitude of causes that include, but are not limited to, injury, inflammation, underlying disease, or idiopathic health conditions. Over time, pain of the chronic variety can drastically diminish an individual’s quality of life via mobility limitation, sleep disturbances, depression, anxiety, and overall reduced productivity.

In the United States, chronic pain is amongst the most commonly reported chronic conditions [2]. In the 2019 National Health Interview Survey, approximately 20.4% of American adults reported having chronic pain that was present every day or on most days for the past three months. Approximately 7.4% of American adults reported having high-impact chronic pain, which was defined as chronic pain that limited their life and work activities every day or on most days for the past three months. 

According to recent studies, chronic pain continues to inflict enormous strain on public health, with total societal costs reaching as high as USD 635 billion a year in the United States alone [3]. The contributors to these costs are multifactorial in nature, ranging from direct medical treatments, such as various pain management modalities and prescription opioids, to refractory opioid dependence, as well as lost productivity [4]. The economic burden that chronic pain places on the United States exceeds the amount spent on the country’s preeminent health conditions of diabetes, cancer, and cardiovascular disease combined [5].

Despite the massive amounts spent on treatment, chronic pain is rarely resolved with total alleviation, as current medical approaches are limited in their efficacy [6]. The crux of this issue is rooted in the concept and nature of pain itself. Pain remains a longstanding and challenging issue to address for healthcare providers worldwide because it is a subjectively perceived sensation with underlying objective, biochemical origins [7]. Further complicating the diagnosis and treatment of chronic pain is the apparent lack of a linear correlation between pain severity and the type or degree of organic pathology [8]. By contrast, each individual’s experience of chronic pain is formed by countless behavioral, biochemical, and psychosocial determinants [9]. As a result, the diagnosis and treatment of chronic pain is heavily reliant on and complicated by subjective ratings, which are often extremely variable due to the myriad of individual differences in pain sensitivity across patients [10]. The inherent biopsychosocial complexity of chronic pain coupled with highly subjective patient self-reporting has ultimately limited the ceiling of efficacy for many treatment methodologies.

## 2. Therapeutic Potential of Chronic Pain Biomarkers

The diagnostic tools currently used to evaluate pain are highly influenced by subjective assessments, inevitably restricting the effectiveness of corresponding treatment modalities. In comparison to other pain therapeutics, opioid analgesics continue to display a relatively high level of effectiveness for treating chronic pain [11]. However, their effectiveness does not come without a price, as it is well-documented that with prolonged use of this class of analgesics, individuals are at a higher risk for developing opioid abuse, addiction, and dependence. The resulting opioid crisis has plagued the United States since the 1990s and continues to worsen each year, with opioid-involved overdose deaths reaching a new high of 80,411 across all ages [12]. Unfortunately, the opioid crisis cannot simply be fixed by removing these analgesics [7]. Opioids are highly addictive, and an abrupt elimination of their use would bring about a slew of difficult problems for the healthcare community to address, making their replacement a particularly challenging dilemma. 

A major contributing factor behind the relative lack of success in finding alternative medications to replace opioids is our somewhat nominal understanding of the biochemical mechanisms that underlie chronic pain. A deeper comprehension of the underlying biochemical mechanisms that govern the subjective experience of chronic pain would open the door to the development of novel and personalized non-opioid treatments that precisely regulate pain pathways [7]. In order to obtain a better grasp of these biochemical mechanisms, researchers have directed large amounts of effort towards identifying biomarkers that are associated with chronic pain or may be potentially causing the onset or exacerbation of it [13]. 

A biomarker is defined as a substance in an organism that can be accurately and reproducibly measured to indicate regular biological processes, pathological biological processes, or pharmacological responses to a treatment [14]. Biomarkers are widely utilized across various medical specialties as an objective metric, providing insight into and guidance for diagnosing diseases, monitoring disease progression, and evaluating drug safety and efficacy during clinical trials [15]. Sadly, identifying biomarkers for chronic pain has been especially challenging considering its biopsychosocial intricacy and the frequent prevalence of comorbid psychiatric illnesses such as anxiety and depression [13]. Previous studies have proposed that specific cerebrospinal fluid, salivary, serum, and urinary biomarkers could be useful in identifying patients at risk of chronic pain development and may serve as prognostic markers for disease progression and treatment efficacy [16]. Despite their semblance of potential, the usefulness of these current individual and panel-based chronic pain biomarkers for guiding clinical decision making remains limited. Due to all these aforementioned reasons, pain biomarker research has expectedly fallen behind other medical specialties as of late. 

A further complicating circumstance in the search for chronic pain biomarkers is the logicality of the search itself [17]. It has been argued by some that discovering biomarkers for chronic pain is simply not plausible given the fact that pain is a subjective experience. Conversely, there is a general consensus amongst researchers that pain will never be a quantifiable condition due to the very nature of its subjectiveness [18]. As a result, chronic pain biomarker research has shifted its primary goal towards identifying biomarkers that correlate with the underlying neurobiological mechanisms behind painful conditions, rather than substances that are merely present during chronic pain states. More specifically, this approach to seeking pain biomarkers aims to facilitate the diagnosis and treatment of chronic pain through the perspective of underlying pathophysiological mechanisms instead of just symptoms [19]. In particular, focusing research efforts towards identifying mechanistic pain biomarkers could introduce a new generation of analgesic medications that possess disease-modifying properties [13].

As of late, a particular mechanism of interest to chronic pain researchers has been the kynurenine pathway (KP) of tryptophan metabolism [20]. Researchers believe the KP may be a promising source of chronic pain biomarkers because of its crucial role in modulating neuroinflammation and neuroplasticity. Studies have found that the activation of this pathway results in the production of metabolites that contain pro-inflammatory and pro-nociceptive effects, which in turn can contribute to the development and persistence of chronic pain [20]. Additionally, disruptions in the kynurenine pathway have also been observed in various chronic pain conditions, suggesting its potential as a biomarker for chronic pain [21]. Hence, targeting the kynurenine pathway may be a promising therapeutic strategy for treating chronic pain, and identifying biomarkers associated with this pathway could lead to the development of more effective pain management approaches. In fact, previous attempts at therapeutic interventions have already been made outside of the realm of strictly chronic pain, targeting KP metabolites to address migraine, neuropathic, and sciatic nerve pain in particular [21]. Furthermore, in prior studies that evaluated KP metabolites as potential pain biomarkers, KP metabolites exhibited greater quantifiability and were more reliably sensitive than their current biomarker counterparts, further reinforcing their potential as a useful biomarker for monitoring and managing chronic pain as well.

## 3. The Kynurenine Pathway: An Overview

Tryptophan (Trp) is an essential amino acid that is metabolized by way of two distinct branches known as the serotonin and kynurenine pathways (KP), illustrated in Figure 1 [22]. Trp is predominantly recognized for being the sole precursor of peripherally and centrally produced serotonin and, subsequently, melatonin via the serotonin pathway [23]. However, only a small fraction of Trp is converted into serotonin. The overwhelming majority of tryptophan is consumed by way of the KP, which is the primary pathway responsible for the catabolism of approximately 99% of ingested tryptophan that is not utilized for protein synthesis [24]. This pathway can be activated by stress and immunocytokines, which can consequently divert available tryptophan away from serotonin production [25]. 

In mammals, the KP is started via the conversion of Trp into *N*-formylkynurenine, which is primarily driven by the enzymes indoleamine 2,3-dioxygenase (IDO) and tryptophan 2,3-dioxygenase (TDO) [26]. *N*-formylkynurenine is subsequently broken down by kynurenine formamidase to produce kynurenine (KYN). KYN can then be further metabolized through three sub-branches. The first branch mainly occurs in microglia and involves the degradation of KYN into 3-hydroxykynurenine (3-HK) and 3-hydroxyanthranilic acid (3-HAA) by kynurenine 3-monooxygenase (KMO) and kynureninase, respectively [27]. 3-HAA can then be converted by 3-hydroxyanthranilic acid 3,4-dioxygenase (3HAO) into α-amino-α-carboxymuconic-ω-semialdehyde [28]. During physiological states, quinolinic acid (QA) is formed from α-amino-α-carboxymuconic-ω-semialdehyde via spontaneous rearrangement [29]. Quinolinate phosphoribosyltransferase (QPRT) then acts as the final rate-limiting enzyme in the portion of the KP that gives rise to the eventual generation of nicotinamide adenine dinucleotide (NAD^+^) [30]. This aforementioned process is recognized as the de novo pathway of NAD^+^ synthesis.

The second branch of the KP also occurs in microglia, but is initiated by kynureninase activity on KYN [31]. Similarly to 3-HK, KYN can also serve as a substrate for kynureninase, resulting in the production of anthranilic acid (AA). In general, the kynureninase found in mammals favors the conversion of 3-HK into 3-HAA [32]. However, in the brain, AA serves as the preferred precursor of 3-HAA in comparison to when it is found in peripheral organs [33]. For reasons unknown to researchers, it appears that the frequency with which kynureninase catalyzes the conversion of KYN into AA, and subsequently 3-HAA, is dependent on the location in the body where its activity is occurring at a given moment in time.

The third branch of the KP involves the synthesis of kynurenic acid (KA) from KYN in peripheral skeletal muscle and astrocytes [34]. KA is created through the irreversible transamination of KYN by kynurenine aminotransferases (KATs) [35]. Four types of KATs have demonstrated the ability to catalyze this transamination, but within the mammalian brain, KAT II is believed to be the foremost biosynthetic enzyme of KA [36]. This is due to the observation that even in the ample presence of competing amino acids, KAT II exhibits a particularly high substrate specificity for KYN, making it the most likely driver of KA synthesis.

Out of the various branches that KYN metabolism is composed of, the two routes that respectively result in the metabolites QA and KA appear to be especially implicated in chronic pain states, with the former metabolite acting as a precursor for the coenzyme NAD^+^, a crucial chemical in cellular energy generation [25]. In addition to being involved in cellular functioning, dysfunction of the KP can also lead to increased levels of metabolites that are neurotoxic themselves or strongly associated with inflammatory and neurodegenerative states, such as 3-HK, 3-HAA, and QA [37]. The concurrent reduction in serotonin production and function has implicated the KP in many psychiatric conditions as well. There is extensive evidence that demonstrates the presence of neurotoxic and inflammatory activity within chronic pain conditions, therefore implicating the KP as a common mechanistic denominator.

## 4. Kynurenine Pathway Metabolite Biomarkers in Chronic Pain

### 4.1. Quinolinic Acid

During physiological states, the favored route of the KP results in the eventual conversion of KYN into NAD^+^, with QA as an intermediate byproduct [38]. QA is a neuroactive metabolite of the KP that can influence the development of pain hypersensitivity and depression [7]. More specifically, QA acts as a potent excitotoxin in the central nervous system, primarily through its ability to act as an *N*-methyl-D-aspartate (NMDA) receptor agonist in the brain [39]. QA also exhibits destructive effects via lipid peroxidation and cytoskeletal destabilization [40].

The vast majority of QA present in the body is produced by microglia and macrophages, hence making QA a particularly sensitive marker of chronic systemic inflammation [41]. Production of QA is greatly increased during an immune response, the mechanism of which is suspected to be via the activation of IDO-1, IDO-2, and TDO by inflammatory cytokines (primarily IFN gamma) [42]. During inflammatory states, cytokine and chemokine production aid in the retention of leukocytes in the brain, resulting in the degradation of the blood–brain barrier (BBB) and permitting more QA to enter the brain [43]. Moreover, QA has also been implicated in the destabilization of astrocyte and brain endothelial cell cytoskeletons, further contributing to the disintegration of the BBB and even higher concentrations of QA in the brain. 

Abnormally high levels of QA can cause hindered neuronal cell function or apoptotic death [44]. When excessive levels of QA are produced during a state of inflammation, the NMDA receptor is overexcited, leading to an influx of calcium (Ca^2+^) into the neuron [45]. Excessive levels of Ca^2+^ activate enzymatic pathways that degenerate critical proteins in the cell and increase intracellular nitric oxide levels, ultimately resulting in an apoptotic response by the neuron. 

Another mechanism of neurotoxicity exhibited by QA is its detrimental effects on the glutamate–glutamine cycle within astrocytes [46]. The glutamate–glutamine cycle is essential for the recycling of glutamate and preventing toxic levels of glutamate from accumulating inside the synapses of astrocytes. At elevated levels, QA inhibits glutamine synthetase, preventing the condensation of glutamate and ammonia into glutamine. This consequently increased level of glutamate further agonizes NMDA receptors, working synergistically with QA to upregulate glutamate levels even more while simultaneously inhibiting glutamate uptake. In essence, QA has the ability to self-potentiate its own toxicity. Additionally, QA can irreparably alter astrocyte structure and biochemistry, triggering apoptosis [47]. Astrocyte apoptosis generates a pro-inflammatory response that exacerbates the original inflammatory response that initiated QA production.

Chronic pain is one of the most common comorbidities of persistent inflammatory states. The substantial involvement of the KP in chronic inflammation mechanisms has consequently made its metabolites compelling targets for biomarker investigation in previous studies. Of the KP metabolites, elevated QA has been found to be the most common abnormal biomarker amongst chronic pain subjects. In a retrospective analysis of 17,834 unique chronic pain samples, elevated QA was observed in 29% of patients, the highest out of eleven compounds in a pain-specific biomarker panel that evaluated essential micronutrients for nerve health (methylmalonic acid, xanthurenic acid, homocysteine, and 3-hydroxypropylmercapturic acid), chronic inflammation (QA and KA), oxidative stress/damage (pyroglutamate, hydroxymethylglutarate, and ethylmalonic acid), and neurotransmitter turnover (5-hydroxyindoleacetate and vanilmandelate) [13]. In a subsequent study of 298 chronic pain patients, the results of Gunn et al., 2020, were confirmed, as QA was once again found to be the most common abnormal biomarker in the study population, observed in 36% of patients [48].

### 4.2. Kynurenic Acid

Conversely, kynurenine that is not used to produce NAD^+^ is converted into kynurenic acid (KA) via kynurenine aminotransferase (KAT) enzymes [49]. KA is also a neuroactive metabolite of the KP, but unlike its QA counterpart, it acts as an antiexcitotoxin and is largely neuroprotective in function [50]. KA’s mechanism of action has been attributed to its activity at various targets, illustrated in Figure 2. 

At elevated micromolar concentrations, KA is a broad-spectrum ionotropic glutamate receptor antagonist at α-amino-3-hydroxy-5-methyl-4-isoxazolepropionic acid (AMPA), NMDA, and Kainate glutamate receptors [51]. However, KA exhibits preferential affinity for the NMDA receptor, consequently directing the majority of its attenuating activity as a competitive inhibitor there [52]. The NMDA receptor is a tetramer structure composed of four subunits [53]. The NR1 and NR2 subunits are contained within the most common varieties of NMDA receptors found in the brain [54]. KA has a particularly high affinity for the obligatory glycine co-agonist (glycine B) site contained on the NR1 subunit, competitively inhibiting the NMDA receptor at low micromolar concentrations (EC_50_ = 7.9 to 15 µM) [55]. KA was also shown to inhibit the glutamate-binding site on the NR2 subunit at higher micromolar concentrations (EC_50_ = 200 to 500 µM) as well. Since the NMDA receptor is the ionotropic glutamate receptor subtype that is most permeable to Ca^2+^, its agonism is often implicated as the main driver of excitotoxic states [56]. Therefore, the neuroprotective effect of KA can be primarily ascribed to its ability to antagonize NMDA receptors, which subsequently inhibits glutamate excitotoxicity.

KA has also demonstrated agonist activity at certain G-protein-coupled receptors (GPCRs) as well [57]. Of particular interest is G-protein-coupled receptor 35 (GPR35), at which KA acts as an endogenous ligand, inhibiting adenylate cyclase activity and N-type Ca^2+^ channels in astrocytes and sympathetic neurons, resulting in decreased intracellular levels of cyclic adenosine monophosphate (cAMP) and Ca^2+^, respectively [58]. Furthermore, the activation of GPR35 via KA has also exhibited inhibitory effects on phosphoinositide 3-kinase (PI3K)/protein kinase B (Akt) and mitogen-activated protein kinase (MAPK) pathways [59]. KA-GPR35 signaling has been shown to decrease the phosphorylation of Akt, extracellular signal-regulated kinase (ERK), and p38 mitogen-activated protein kinase (p38), while also inducing the accumulation of β-catenin. Since the PI3K/Akt, MAPK, and β-catenin pathways are widely recognized targets of GPR signaling, it is plausible that the observed inhibition of ERK and p38, along with the induction of β-catenin accumulation, are a result of KA-mediated GPR35 activation [60].

With the aforementioned effects of KA and GPR35 interaction in mind, it is possible that KA-GPR35 signaling may indirectly lead to the attenuation of the inflammatory response [58]. In a previous study by Elizei et al., the IL-23/IL-17 immune axis, an important player in the development of chronic inflammation, was found to be downregulated following KA administration [61]. Considering the cAMP signaling cascade’s well-known involvement in the regulation of innate and adaptive cell functions, there is compelling reason to believe that the downregulation of the IL-23/IL-17 immune axis is due to KA-GPR35-mediated inhibition of adenylate cyclase [61]. With regard to intracellular Ca^2+^ concentration, elevated levels correlate with the activation of crucial inflammatory transcription factors such as NF-κB [62]. Therefore, it can be posited that KA alternatively reduces inflammation by way of KA-GPR35-mediated inhibition of N-type Ca^2+^ channels, which in turn reduces inflammatory signal secretion. Similarly, KA may also interfere with inflammation generation via KA-GPR35-mediated inhibition of the PI3K/Akt and MAPK pathways, both of which are essential for producing an inflammatory response [63,64]. In contrast, the β-catenin signaling pathway is recognized to inhibit inflammation via the stabilization of NF-κB inhibitory IκB-factors, which consequently limits NF-κB activation [65]. KA-GPR35-mediated induction of β-catenin accumulation provides evidence of yet another mechanism of action by which KA can inhibit inflammation [66]. See Table 1 for a summary of these findings.

As previously discussed, persistent inflammatory states are highly correlated with chronic pain. In patients who experience comorbid pain with various psychiatric conditions, diminished levels of neuroprotective metabolites have been frequently identified [34]. KA’s functions as a neuroprotective metabolite and downregulator of the inflammatory response suggest that these patients’ chronic pain symptoms could potentially be attributed in part to diminished levels of KA and other metabolites with similar properties. Thus, decreased levels of KA could be utilized as a reliable biomarker for indicating chronic pain states undergoing poor inflammatory regulation. Conversely, elevated levels of KA could also correlate with an initial upregulation of the KP in response to chronic pain disorders. In a retrospective observational study by Pope et al., a sample size of 298 chronic pain patients were given the patient-reported outcomes measurement information system (PROMIS-29) survey, along with a urine test, before the start of therapy [48]. The PROMIS-29 survey was used to evaluate eight universal domains that include anxiety, depression, fatigue, sleep disturbance, pain interference, pain intensity, physical function, and ability to participate in social roles and activities. Meanwhile, the urine test was used to assess pain biomarkers and their respective relationships to the domains of interest from the PROMIS-29 survey. In 33% of these chronic pain patients, elevated KA levels were found and KA’s increased presence was strongly associated with pain interference. Some researchers have also hypothesized that during instances of neurological degeneration, elevated levels of KA are observed as a result of unsuccessful efforts to protect cells [67]. Whether it is increased or decreased KA levels that are observed, it is evident that KA is highly implicated in inflammatory and chronic pain states.

### 4.3. Kynurenic Acid/Quinolinic Acid Ratio

Since KA and QA generation respectively occur within diverting branches of the KP, circulating levels of KA and QA can often share an inverse relationship to each other [25]. In addition to this anticorrelation of production, the opposing properties of neuroprotective KA and neurotoxic QA add another layer of complexity to their relationship. Imbalances between these two classes of metabolites are of particular interest, as researchers have speculated that comorbid chronic pain in patients with depression could be attributed to a disproportionate presence of neurotoxins without reciprocal neuroprotectant response [34]. A 2021 study by Groven et. al. showed that patients with chronic fatigue syndrome (CFS) had reduced KA/QA ratios in comparison to healthy controls [68]. This observation lends credence to the theory that these patients’ symptoms may be caused by a deficiency in the neuroprotective metabolites available to combat elevated neurotoxin levels. In future studies, this notion could be quantitatively assessed and corroborated by the KA/QA ratio or another associated neuroprotective/neurotoxic KP metabolite ratio, supporting the idea that these ratios are potentially useful biomarkers for chronic pain conditions.

### 4.4. Kynurenine/Tryptophan Ratio

As per the earlier overview of the KP, it is well recognized that IDO is one of the primary enzymes that drives the conversion of Trp to KYN. In accordance with this characteristic, prior clinical studies have discovered a correlation between elevated IDO levels and the presence of chronic, inflammation-based pain. In a study carried out by Barjandi et al. 2019, a sample size of 113 female patients was examined, 40 of which had fibromyalgia, 17 with temporomandibular disorders myalgia, and 56 were healthy controls [69]. For all study participants, KYN and Trp plasma levels were measured via enzyme-linked immunosorbent assay (ELISA) testing. Their findings revealed a statistically discernible inverse relationship between pain intensity and Trp plasma levels (*p* < 0.001), indicating that the greater the pain intensity is, the lower the Trp plasma levels are. Conversely, the results also showed a statistically discernible positive relationship between pain intensity and the kynurenine/Trp ratio (*p* < 0.001), indicating that elevated pain correlates with increased kynurenine levels. In another study by Staats Pires et al. 2020, individuals with chronic back pain had considerably higher IDO levels and kynurenine/Trp ratios than their healthy counterparts [70]. Together, these associations lend support to the notion that in circumstances of inflammation-induced pain, the majority of Trp is metabolized via the KP.

As previously discussed, it is important to note that Trp acts as the precursor for the neurotransmitter serotonin [71]. Since IDO is responsible for the initiation of the KP, elevated levels of the enzyme consequently demonstrate a negative correlation with serotonin levels as Trp is shunted away from the serotonin pathway and towards the KP instead. For that reason, during inflammatory conditions where IDO exhibits sustained consumption of Trp, abnormally low levels of serotonin are detected because there is reduced Trp substrate available for serotonin synthesis. A decreased serotonin/Trp ratio concurrently present with an increased kynurenine/Trp ratio has also been observed in model rats with chronic arthritis inflammatory pain, further validating the mechanism behind the relationship between these two ratios [70].

**Table 1 pharmaceuticals-16-00681-t001:** Summary of kynurenine pathway metabolites and metabolite ratios as potential biomarkers in chronic pain.

Metabolite	Clinical Significance	Chronic Pain Implications	Mechanism
Quinolinic acid (QA)		Hyperalgesia development Comorbid chronic pain with neurodegenerative and psychiatric disorders	Excitotoxin via NMDA receptor agonism [39]
	Cytokine-mediated chronic inflammation		Self-potentiation of neurotoxicity via interference of glutamate–glutamine cycle [46]
			Elevated levels involved in neuronal cytoskeleton destabilization and apoptosis [40]
Kynurenic acid (KA)	Cytokine-mediated chronic inflammation	Neuroprotective properties Diminished levels possibly indicate poor inflammatory regulation with subsequent pain exacerbation Elevated levels possibly indicate upregulation for initial response to pain	Anti-excitotoxin via noncompetitive NMDA receptor antagonism [51]
			Anti-inflammatory via GPR35-mediated agonism [58] Downregulation of PI3K/Akt and MAPK pathways (inflammatory) [59] Upregulation of β-catenin accumulation (anti-inflammatory) [66]
Kynurenic acid/Quinolinic acid ratio (KA/QA)	Cytokine-mediated chronic inflammation Neurotoxicity	Lower KA/QA ratios indicate a lack of neuroprotection with subsequent pain exacerbation	Inadequate neuroprotective response via overactivity of QA production relative to KA production [68]
Kynurenine/Tryptophan ratio (KYN/Trp)	Cytokine-mediated chronic inflammation	Higher KYN/Trp ratios indicate elevated pain intensity	Upregulated IDO levels shunt available Trp away from serotonin production and towards the KP [70]

## 5. Conclusions

In addition to enhancing our comprehension and our ability to more accurately diagnose chronic pain conditions, the successful identification of mechanistic chronic pain biomarkers would also facilitate the development of novel, personalized, non-opioid therapies that can act as disease-modifying agents. Biomarkers would also grant researchers and healthcare professionals the ability to monitor the effectiveness of these novel modulating treatments over extended periods of time. 

Objective biomarkers are foundational to the practice of personalized medicine, and identifying and verifying biomarkers to aid in the diagnosis and treatment of chronic pain conditions would substantially lower healthcare expenses worldwide. While the discovery of any pain-specific biomarker is a meaningful contribution to our continuous pursuit towards further understanding the pathophysiology of chronic pain, efforts must be geared towards the most critical and influential biomarkers: those that can be modified to alter the course of a disease.

## Figures and Tables

**Figure 1 pharmaceuticals-16-00681-f001:**
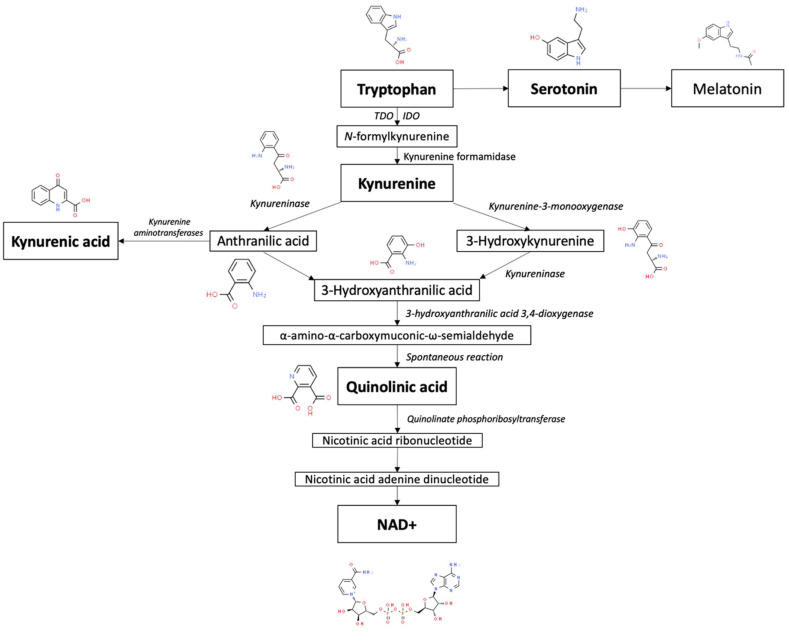
Overview of tryptophan metabolism via the kynurenine pathway. Chemical pictures from the Royal Society of Chemistry. www.chemspider.com. Accessed on 20 April 2023.

**Figure 2 pharmaceuticals-16-00681-f002:**
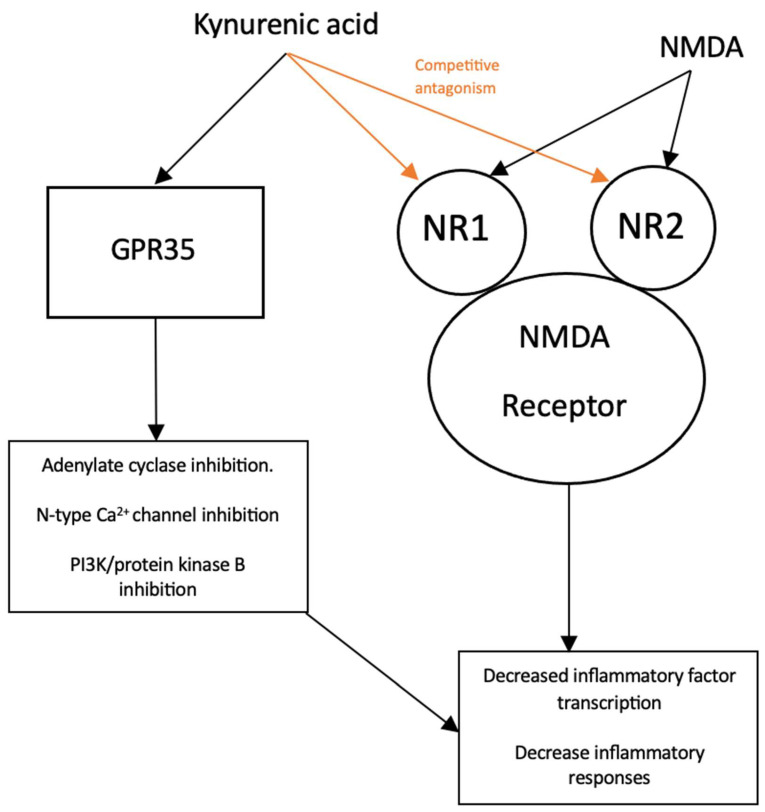
Simplified schematic of kynurenic acid and its effects on reducing inflammatory mediators.

## Data Availability

Data is contained within the article.

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
