# Peer review of "Kynurenine Pathway Metabolites as Potential Biomarkers in Chronic Pain"

_pharmaceuticals, 2023, doi:10.3390/ph16050681_

Round 1
Reviewer 1 Report
Chronic pain leads to significantly decreased quality of life, reduced productivity, worsening of chronic disease, and psychiatric disorders such as depression, anxiety, and substance abuse disorders. On the other hand, some metabolites of kynurenine pathway seems to be an important mediator in neuropathic pain pathology and have also been implicated in several inflammation-associated pain syndromes and depressive mood disorders. Based on these, the literature review of the subject area can be considered justified. According to the authors' concept, dysregulation of the kynurenine pathway and changes in the ratios of its metabolites have been associated with numerous neurotoxic and inflammatory states, many of which present simultaneously with chronic pain symptoms. Moreover, levels of Quinolinic Acid and/or Kynurenic Acid, as well as ratios of Kynurenine/Tryptophan and Kynurenic Acid/Quinolinic Acid can act individually and together as indicative biomarkers during chronic pain. According to the reviewer, the article is suitable for publication. The appearance of the first figure was shadowy, this needs to be corrected.
Author Response
- Reviewer's comment: “Chronic pain leads to significantly decreased quality of life, reduced productivity, worsening of chronic disease, and psychiatric disorders such as depression, anxiety, and substance abuse disorders. On the other hand, some metabolites of kynurenine pathway seems to be an important mediator in neuropathic pain pathology and have also been implicated in several inflammation-associated pain syndromes and depressive mood disorders. Based on these, the literature review of the subject area can be considered justified. According to the authors' concept, dysregulation of the kynurenine pathway and changes in the ratios of its metabolites have been associated with numerous neurotoxic and inflammatory states, many of which present simultaneously with chronic pain symptoms. Moreover, levels of Quinolinic Acid and/or Kynurenic Acid, as well as ratios of Kynurenine/Tryptophan and Kynurenic Acid/Quinolinic Acid can act individually and together as indicative biomarkers during chronic pain. According to the reviewer, the article is suitable for publication
Response: Thank you very much for the kind words and compliments on the revised manuscript.
- Reviewer's comment: The appearance of the first figure was shadowy, this needs to be corrected.
Response: Thank you for the suggestion. The shadowing of the figure has been resolved.
Reviewer 2 Report
Chronic pain is a significant and costly problem in the United States as well as throughout the industrialized world. Unfortunately, there have been concerns about the effectiveness of traditional medical interventions, suggesting the need for alternative chronic pain treatment strategies. Therefore, the presented manuscript is timely and scientifically sound and properly written, following all the guidelines for publications of this tape of publications. The figure and table are an interesting collection of available informations. Conclusions correctly summarize the manuscript. Literature is correctly cited and up to date. I suggest only the inclusion of more information about what other biomarkers are currently known and what is the difference between them and the Kynurenine Pathway Metabolites. Please also explain why the Authors are interested in this pathway and how the available information on its role in pain can be used (diagnostic methods, potential drugs).
Author Response
Reviewer's comment: “Chronic pain is a significant and costly problem in the United States as well as throughout the industrialized world. Unfortunately, there have been concerns about the effectiveness of traditional medical interventions, suggesting the need for alternative chronic pain treatment strategies. Therefore, the presented manuscript is timely and scientifically sound and properly written, following all the guidelines for publications of this tape of publications. The figure and table are an interesting collection of available informations. Conclusions correctly summarize the manuscript. Literature is correctly cited and up to date.”
Response: Thank you very much for the kind words and compliments on the revised manuscript.
- Reviewer's comment: “I suggest only the inclusion of more information about what other biomarkers are currently known and what is the difference between them and the Kynurenine Pathway Metabolites.”
Response: Thank you for the suggestion. We have included details regarding currently known biomarkers and their presently limited utility in assisting with clinical decision making. We have also addressed that the differences that set kynurenine pathway metabolites apart from current biomarkers are their greater potential for quantifiability and more reliable sensitivity.
“Previous studies have proposed that specific cerebrospinal fluid, salivary, serum, and urinary biomarkers could be useful in identifying patients at risk of chronic pain development and may serve as prognostic markers for disease progression and treatment efficacy [16]. Despite their semblance of potential, the usefulness of current individual and panel-based chronic pain biomarkers for guiding clinical decision-making remains limited.”
“Furthermore, in prior studies that evaluated KP metabolites as potential pain biomarkers, KP metabolites exhibited greater quantifiability and were more reliably sensitive than its current biomarker counterparts, further reinforcing its potential as a useful biomarker for monitoring and managing chronic pain as well.”
- Reviewer's comment: “Please also explain why the Authors are interested in this pathway and how the available information on its role in pain can be used (diagnostic methods, potential drugs).”
Response: Thank you for the suggestion. We are interested in the kynurenine pathway and its metabolites because they have already shown prior potential as therapeutic targets in the treatment of other subtypes of pain (neuropathic, migraine, and sciatic pain). Kynurenine metabolites have also exhibited a great degree of quantifiability and reliable sensitivity, further supporting its compelling potential for monitoring and managing chronic pain. We have included two sentence to reflect these sentiments.
“In fact, previous attempts at therapeutic interventions have already been made outside of the realm of strictly chronic pain, targeting KP metabolites to address migraine, neuropathic, and sciatic nerve pain in particular [21].”
“Furthermore, in prior studies that evaluated KP metabolites as potential pain biomarkers, KP metabolites exhibited greater quantifiability and were more reliably sensitive than its current biomarker counterparts, further reinforcing its potential as a useful biomarker for monitoring and managing chronic pain as well.”
Reviewer 3 Report
The manuscript „Kynurenine Pathway Metabolites as Potential Biomarkers in Chronic Pain” approaches an important problem of chronic pain biomarkers. Kynurenine pathway gained significant interest recently, as evidenced by several review papers. The application of the pathway steps as biomarkers and potential therapeutic targets is mentioned – some examples of therapeutic interventions (or attempts) would be important for this text. The practical aspect of biomarkers is associated with the stability of selected compounds, and sensitivity of detection methods in relation to the physiological levels. Comments on this aspect of KP would be of interest to readers (only ELISA and urine test are mentioned in the text).
Pain-specific biomarker panels (mentioned, for example, in line 216) are aimed at several compounds. Examples of monitored substances and the benefits of KP metabolites should be discussed.
The tryptophan metabolism presented in Figure 1 would be improved by introduction of structural formulas of the mentioned compounds.
A scheme for KA activities (summarizing the 4.2 part) would be really interesting.
Some examples of neurotoxic metabolites would be a good addition to sentence in lines 168-169.
Table 1 needs references for mentioned mechanisms, as it attracts readers to this part of the text.
The effect of QA leading to apoptosis is mentioned twice (lines 194-199 and 208-209).
Please clarify the sentence “Since the NMDA receptor is the ionotropic glutamate receptor subtype that is most permeable to Ca2+, its 237 agonism is often implicated as the main driver for [54].” (line 236)
Please clarify “longitudinally monitor” in line 345
There are formal problems concerning presentation of chemical symbols and names (for example: line 132 N-formylkynurenine (N in italics), line 182 N-methyl-D-aspartate (N in italics, D in smaller font), ions: NAD+ (line 144 and following) and Ca2+ (line 197 and following) with (+, 2+) in superscript as ion charge). EC50 is usually presented as EC50 (line 234).
The references used in the text are sometimes located in the first sentence of paragraph, with detailed data following in longer text without reference, creating doubt about sources (see paragraph containing ref 19, from line 109). In some cases, the second sentence seems to be a re-phrasing of the first (for example lines 213-216: “Of the KP metabolites, elevated QA has been found to be the most common abnormal biomarker finding amongst chronic pain subjects [13]. In a retrospective analysis of 17,834 unique chronic pain samples, elevated QA was observed in 29% of patients, the highest of the pain-specific biomarker panel” – do the data come from the same reference?).
Some sources should be reconsidered, as there are references to materials containing references to original work (second-hand sources, see ref. 5). Did the Authors use original text in Swedish for ref. 17? (the name of the Author (Emmanuel Bäckryd) contains special character that is missing in the reference).
The text is presented in an impressive language style, although some sentences seem overly complex (for example lines 107-109, line 292).
Although the term catalyzation is used in literature, it looks artificial here, limited to one transformation (lines 138 and 150).
There are few minor issues:
Line 241: KA has also demonstrated agonist activity at certain G-protein-coupled receptors 241 (GPCRs) as well [55].
Line 298: the opposing properties of neuroprotective KA and neurotoxic QA adds another layer of complexity.
Author Response
Reviewer #3
- Reviewer's comment: “The application of the pathway steps as biomarkers and potential therapeutic targets is mentioned – some examples of therapeutic interventions (or attempts) would be important for this text.”
Response: Thank you for the suggestion. We have included some examples of therapeutic attempts that have previously targeted kynurenine pathway metabolites to address various subtypes of pain.
“In fact, previous attempts at therapeutic interventions have already been made outside of the realm of strictly chronic pain, targeting KP metabolites to address migraine, neuropathic, and sciatic nerve pain in particular [21].”
- Reviewer's comment: “The practical aspect of biomarkers is associated with the stability of selected compounds, and sensitivity of detection methods in relation to the physiological levels. Comments on this aspect of KP would be of interest to readers (only ELISA and urine test are mentioned in the text)”
Response: Thank you for your suggestion. We have included a sentence that addresses the advantages of kynurenine pathway metabolites as biomarkers compared to current pain biomarkers, in particular their greater quantifiability and more reliable sensitivity. Unfortunately, we were unable to explore other testing modalities, as only ELISA and urine testing have been adequately researched with respect to detecting kynurenine pathway metabolites. We do agree that further exploration of other testing modalities would be a valuable pursuit.
“Furthermore, in prior studies that evaluated KP metabolites as potential pain biomarkers, KP metabolites exhibited greater quantifiability and were more reliably sensitive than its current biomarker counterparts, further reinforcing its potential as a useful biomarker for monitoring and managing chronic pain as well.”
- Reviewer's comment: “Pain-specific biomarker panels (mentioned, for example, in line 216) are aimed at several compounds. Examples of monitored substances and the benefits of KP metabolites should be discussed.”
Response: Thank you for the suggestion. We have included an additional sentence to address the benefits of KP metabolites as targets for biomarker investigation. We have also edited the sentence that discusses the pain-specific biomarker panel to include specific examples of the monitored substances and the processes they evaluate as well.
“The substantial involvement of the KP in chronic inflammation mechanisms have consequently made its metabolites compelling targets for biomarker investigation by previous studies.”
“In a retrospective analysis of 17,834 unique chronic pain samples, elevated QA was observed in 29% of patients, the highest out of eleven compounds in a pain-specific biomarker panel that evaluated essential micronutrients for nerve health (methylmalonic acid, xanthurenic acid, homocysteine, 3-hydroxypropylmercapturic acid), chronic inflammation (QA, KA), oxidative stress/damage (pyroglutamate, hydroxymethylglutarate, ethylmalonic acid), and neurotransmitter turnover (5-hydroxyindoleacetate, vanilmandelate)”
- Reviewer's comment: “The tryptophan metabolism presented in Figure 1 would be improved by introduction of structural formulas of the mentioned compounds.”
Response: Thank you for the suggestion. We have added structural formulas to highlight the most important of the mentioned compounds in Figure 1.
- Reviewer's comment: “A scheme for KA activities (summarizing the 4.2 part) would be really interesting.”
Response: Thank you for the suggestion. We have included a new figure that provides a scheme for KA activities.
- Reviewer's comment: “Some examples of neurotoxic metabolites would be a good addition to sentence in lines 168-169.”
Response: Thank you for the suggestion. We have included some examples of neurotoxic metabolites to that sentence.
“In addition to being involved in cellular functioning, dysfunction of the KP can also lead to increased levels of metabolites that are neurotoxic themselves or strongly associated with inflammatory and neurodegenerative states, such as 3-HK, 3-HAA, and QA”
- Reviewer's comment: “Table 1 needs references for mentioned mechanisms, as it attracts readers to this part of the text.”
Response: Thank you for the suggestion. We have added the corresponding references alongside the mentioned mechanisms in Table 1.
- Reviewer's comment: “The effect of QA leading to apoptosis is mentioned twice (lines 194-199 and 208-209).”
Response: Thank you for the comment. We wanted to make a distinction between QA mediated apoptosis for both astrocytes and neurons in order to demonstrate its toxic effects on different cells of the CNS, thus highlighting QA’s role in the propagation of chronic pain and inflammatory states.
- Reviewer's comment: “Please clarify the sentence “Since the NMDA receptor is the ionotropic glutamate receptor subtype that is most permeable to Ca2+, its 237 agonism is often implicated as the main driver for [54].” (line 236)”
Response: Thank you for pointing out that incomplete sentence. We have included the missing portion of that sentence to hopefully make it more clear.
“Since the NMDA receptor is the ionotropic glutamate receptor subtype that is most permeable to Ca2+, its agonism is often implicated as the main driver for excitotoxic states.”
- Reviewer's comment: “Please clarify “longitudinally monitor” in line 345”
Response: Thank you for the comment. We have edited the sentence to hopefully make it more clear.
“Biomarkers would also grant researchers and healthcare professionals the ability to monitor the effectiveness of these novel modulating treatments over extended periods of time.”
- Reviewer's comment: There are formal problems concerning presentation of chemical symbols and names (for example: line 132 N-formylkynurenine (N in italics), line 182 N-methyl-D-aspartate (N in italics, D in smaller font), ions: NAD+ (line 144 and following) and Ca2+ (line 197 and following) with (+, 2+) in superscript as ion charge). EC50 is usually presented as EC50 (line 234).”
Response: Thank you for pointing out those formatting issues. We have made the appropriate formatting changes.
- Reviewer's comment: “The references used in the text are sometimes located in the first sentence of paragraph, with detailed data following in longer text without reference, creating doubt about sources (see paragraph containing ref 19, from line 109). In some cases, the second sentence seems to be a re-phrasing of the first (for example lines 213-216: “Of the KP metabolites, elevated QA has been found to be the most common abnormal biomarker finding amongst chronic pain subjects [13]. In a retrospective analysis of 17,834 unique chronic pain samples, elevated QA was observed in 29% of patients, the highest of the pain-specific biomarker panel” – do the data come from the same reference?).”
Response: Thank you for pointing out these discrepancies. We have made the appropriate changes to the mentioned references to hopefully make it more clear exactly which sources are being used.
- Reviewer's comment: “Some sources should be reconsidered, as there are references to materials containing references to original work (second-hand sources, see ref. 5).
Response: Thank you for the suggestion. We have modified a few of our sources to reference the original work.
- Reviewer's comment: Did the Authors use original text in Swedish for ref. 17? (the name of the Author (Emmanuel Bäckryd) contains special character that is missing in the reference).”
Response: Thank you for pointing out this issue and the missing special character. We have replaced that source with a different source from Bäckryd written in English that we feel addresses line 112 even better. We have also included the “ä” in Bäckryd in our reference list.
- Reviewer's comment: “The text is presented in an impressive language style, although some sentences seem overly complex (for example lines 107-109, line 292).”
Response: Thank you for the suggestion. We have made changes to the mentioned sentences to make them less complex and more clear.
“In particular, focusing research efforts towards identifying mechanistic pain biomarkers could introduce a new generation of analgesic medications that possess disease-modifying properties.”
“Whether it is increased or decreased KA levels that are observed, it is evident that KA is highly implicated in inflammatory and chronic pain states.”
- Reviewer's comment: “Although the term catalyzation is used in literature, it looks artificial here, limited to one transformation (lines 138 and 150).”
Response: Thank you for the comment. We have removed the use of catalyzation and edited the sentences accordingly.
“The first branch mainly occurs in microglia and involves the degradation of KYN into 3-hydroxykynurenine (3-HK) and 3-hydroxyanthranilic acid (3-HAA) by kynurenine 3-monooxygenase (KMO) and kynureninase, respectively”
“In general, the kynurenianse found in mammals favors the conversion of 3-HK into 3-HAA.”
- Reviewer's comment: Line 241: KA has also demonstrated agonist activity at certain G-protein-coupled receptors 241 (GPCRs) as well [55].
Response: We have made changes to the sentence to hopefully make it more clear.
“KA has also demonstrated agonist activity at certain G-protein-coupled receptors (GPCRs) as well.”
- Reviewer's comment: Line 298: the opposing properties of neuroprotective KA and neurotoxic QA adds another layer of complexity.
Response:
We have made changes to the sentence to hopefully make it more clear.
“In addition to this anticorrelation of production, the opposing properties of neuroprotective KA and neurotoxic QA adds another layer of complexity.”